# Reducing Variance in Meta-Learning via Laplace Approximation for Regression Tasks

**Alfredo Reichlin**\*  *alfrei@kth.se*
*KTH Royal Institute of Technology*

**Gustaf Tegnér**\*  *gustafte@kth.se*
*KTH Royal Institute of Technology*

**Miguel Vasco**  *miguelsv@kth.se*
*KTH Royal Institute of Technology*

**Hang Yin**  *hayi@di.ku.dk*
*University of Copenhagen*

**Mårten Björkman**  *celle@kth.se*
*KTH Royal Institute of Technology*

**Danica Kragic**  *dani@kth.se*
*KTH Royal Institute of Technology*

**Reviewed on OpenReview:** *https: // openreview. net/ forum? id= Uc2mqNPkEq*

## Abstract

Given a finite set of sample points, meta-learning algorithms aim to learn an optimal adaptation strategy for new, unseen tasks. Often, this data can be ambiguous as it might belong to different tasks concurrently. This is particularly the case in meta-regression tasks. In such cases, the estimated adaptation strategy is subject to high variance due to the limited amount of support data for each task, which often leads to sub-optimal generalization performance. In this work, we address the problem of variance reduction in gradient-based meta-learning and formalize the class of problems prone to this, a condition we refer to as *task overlap*. Specifically, we propose a novel approach that reduces the variance of the gradient estimate by weighing each support point individually by the variance of its posterior over the parameters. To estimate the posterior, we utilize the Laplace approximation, which allows us to express the variance in terms of the curvature of the loss landscape of our meta-learner. Experimental results demonstrate the effectiveness of the proposed method and highlight the importance of variance reduction in meta-learning.

## 1 Introduction

Meta-learning, also known as learning-to-learn, is concerned with the development of intelligent agents capable of adapting to changing conditions in the environment. The central idea of meta-learning is to learn a prior over a distribution of similar learning tasks, enabling fast adaptation to novel tasks given only a limited number of data points. This approach has proven successful in many domains, such as few-shot learning (Snell et al., 2017), image completion (Garnelo et al., 2018b), and imitation learning tasks (Finn et al., 2017b).

One instance of this is Gradient-Based Meta-Learning (GBML), which was introduced in Finn et al. (2017a). GBML methods employ a bi-level optimization procedure, where the learner first adapts its parameters

---

\*Equal contribution.

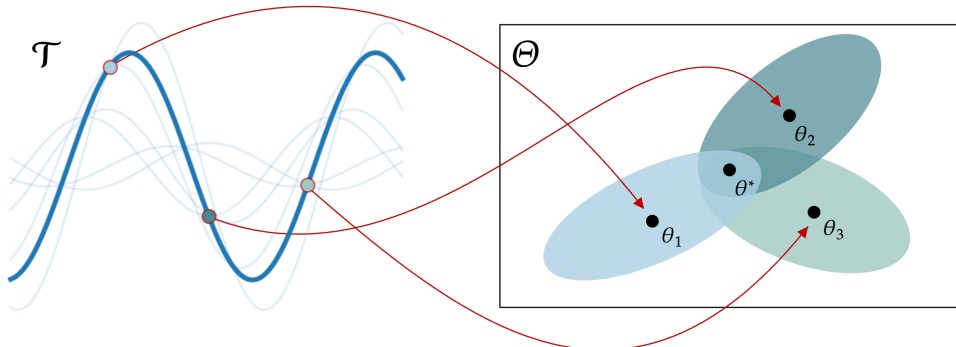

Figure 1: **The Problem of Task Overlap in Regression Tasks**: *On the left*: Three support points of a single task are marked out. These points are shared between different tasks which are marked out by the opaque curves. *On the right:* Each support point induces a distribution in the parameter space. The true function parameters $\theta^*$ lie at the *intersection* over the possible function values.

given a few samples denoted the *support-set*, which it then evaluates on another, more complete, *query-set*. The shared prior is defined as the parameter initialization. A central issue arises in identifying accurate parameters from the support set. The limited amount of support points used for adaptation, measurement error, or ambiguity between the tasks inevitably leads to uncertainty in this parameter identification. To enable more efficient adaptation, more effective priors must be learned.

When learning over a continuous space of tasks in the context of meta-regression, ambiguity often arises. Consider the problem of regressing sinusoidal waves with varying amplitudes and phases as depicted in fig. 1 on the left. Each pair of $(x, y)$ used to adapt to the specific wave can correspond to infinite other waves and, as such, doesn't uniquely identify the task at hand. We refer to this condition as *task overlap*. In this case, the aggregation of the information conveyed by each data point in the adaptation dataset has to consider this uncertainty. GBML models struggle in this regard as this uncertainty is not taken into account.

In this work, we propose **L**aplace **A**pproximation for **V**ariance-reduced **A**daptation **(LAVA)**, a method that introduces a novel strategy for aggregating the information of the support-set in the adaptation process. The key idea of our method is to recognize each support point as inducing a unique posterior distribution over the task parameters (fig. 1 on the right). The complete posterior distribution can then be expressed as the joint posterior over the samples. Thus, our model is a form of Bayesian model-averaging (Hoeting et al., 1999), where each model is induced by a single point from the support. We approximate the posterior distribution w.r.t each support point through *Laplace's approximation*, which constructs a Gaussian distribution where the variance of the estimator is a function of the Hessian of the negative log-likelihood (MacKay, 2003, Chapter 27). In turn, the joint posterior is again Gaussian, from which the optimal value can be expressed as its mean. In contrast to other Bayesian GBML methods, the posterior approximation is not built on the full posterior but rather on the single posteriors induced by every point in the support data. This allows us to optimally aggregate the information these points share and reduce the variance of this estimate in the case of ambiguity.

Our contributions include an insight into the adaptation process of GBML methods by identifying a class of continuous regression problems where GBML methods are particularly subject to high variance in their adaptation procedure. We introduce a method for modeling the variance that each data point carries over the parameters and optimally aggregate these posterior distributions into the task-adapted parameters. Finally, we demonstrate the performance of our method on regression of dynamical systems and real-world experiments and showcase state-of-the-art performance compared to standard GBML methods.

## 2 Preliminaries

### 2.1 Problem Formulation

Let $p(\mathcal{T})$ be a distribution of tasks that share a common generative process. Each task $\tau \in \mathcal{T}$ defines a function $f_\tau : \mathcal{X} \to \mathcal{Y}$ represented as subsets $\tau \in \mathcal{X} \times \mathcal{Y}$ of inputs $\mathcal{X} \subseteq \mathbb{R}^d$ and outputs $\mathcal{Y} \subseteq \mathbb{R}^k$. For each task,

assume we are given a finite dataset sampled i.i.d. i.e., $D_\tau \subseteq \mathcal{X} \times \mathcal{Y}$. These points can be further divided into two sets, the support set $D_\tau^S$ and the query set $D_\tau^Q$ such that $D_\tau = D_\tau^S \cup D_\tau^Q$. We denote $N = |D_\tau^S|$, $M = |D_\tau^Q|$ as the size of the support and query set respectively, which we assume, for simplicity, is the same across all tasks.

From the support set, $D_\tau^S$, we are interested in estimating the underlying function defined by each task, $f_\tau$. Following previous work, (Andrychowicz et al., 2016; Ravi & Larochelle, 2016), this can be described as estimating a map from the support data to a set of parameters that can be used to approximate the true task's function i.e., $\mathcal{A} : \mathcal{X} \times \mathcal{Y} \to \Theta$ with $\mathcal{A}(D_\tau^S) = \theta_\tau$ such that $f_{\theta_\tau} \approx f_\tau$. We define similarity in terms of mean-squared error and compute it using the $M$ points of the query set with the following loss function:

$$\mathcal{L}(\theta_\tau, D_\tau^Q) = \frac{1}{M} \sum_{i=1}^{M} \left\| f_{\theta_\tau}\left(x_\tau^i\right) - y_\tau^i \right\|_2^2, \quad \text{s.t.} \quad \theta_\tau = \mathcal{A}(D_\tau^S). \tag{1}$$

The performance is tightly bound to the error in the estimator $\mathcal{A}$ which in part arises from uncertainty in the support data. In this work, we consider the uncertainty that is induced when data points can belong to multiple tasks, which we denote as *task overlap*.

For ease of notation, let $f_\tau(x) = f(\tau, x)$ denote the true data-generating process.

**Definition 1** (Task Overlap). *We define task overlap as the condition for which $\forall x \in \mathcal{X}$ the map $f(\cdot, x) : \mathcal{T} \to \mathcal{Y}$ is non-injective.*

From a probabilistic perspective, this property is equivalent in stating that the conditional task-distribution $p(\tau \mid x, y)$ will have a non-zero covariance for all $(x, y) \in \mathcal{X} \times \mathcal{Y}$.

An example of *task overlap* can be seen by considering a sine wave regression problem. Let the distribution of tasks be sinusoidal waves with changing amplitude and phase i.e., $y = A_\tau \sin(x + \phi_\tau)$. In this case, the space of tasks is defined by the union of possible amplitudes and phases and thus has dimension 2. A single sample in the form of a tuple $(x, y)$ is, however, insufficient to identify this task unambiguously. In fact, there exists an infinite number of viable amplitudes and phases for which the sine can pass through such a point, fig. 1 on the left. In the terms above, there is no injective map between $(x, y)$ and $(A_\tau, \phi_\tau)$. The set of all possible solutions induced by this single point defines a distribution of solutions in the task-adapted parameter's space. All of these solutions are equally probable if conditioned on this single point. When multiple (and different) samples are considered for the adaptation step, the inference of the sine wave can be exact. There exists a unique sine wave that passes through all of these points at the same time. In terms of task parameters, this can be seen as considering the intersection of the solutions induced by each point; see fig. 1 on the right. When using a single-point estimate, the information about this distribution of possible optimal task parameters is lost.

## 2.2 Gradient-Based Meta-Learning

A notable family of methods to solve the problem described in eq. (1) are GBML algorithms. In these, learning the adaptation process is formulated as a bi-level optimization procedure. A set of meta-parameters $\theta_0$ is learned such that the inference process $\mathcal{A}$ corresponds to a single gradient descent step on the loss computed using the support data and $\theta_0$:

$$\theta_\tau = \theta_0 - \alpha \nabla_\theta \mathcal{L}(\theta, D_\tau^S)\big|_{\theta=\theta_0}, \tag{2}$$

where $\alpha$ is a scalar value denoting the learning rate. The resulting estimate of the task-adapted parameters is then optimized to approximate the underlying task function using eq. (1). This, effectively, defines an overall optimization procedure on the meta-parameters $\theta_0$. Gradient-based Meta-Learning does not require additional parameters for the adaption procedure, as it operates in the same parameter space as the learner. Moreover, it is proven to be a universal function approximator (Finn & Levine, 2018), making it one of the most common models for meta-learning.

As noted in previous work (Grant et al., 2018), the bi-level formulation of GBML can be interpreted from a Bayesian view. We are interested, for each task, in finding a set of adapted parameters that maximize

the likelihood of the query data i.e., $\max p(D_\tau^Q \mid \theta_\tau)$. This is achieved by estimating a prior in the form of $\theta_0$ such that, when combined with the evidence from the support data, it leads to the highest probability estimate. GBML simplifies the process by defining the estimate of the adapted parameters as the solution to eq. (2). This, in fact, is equivalent to estimating the most probable set of parameters given the prior $\theta_0$ and the support data $D_\tau^S$:

$$\hat{\theta}_\tau = \arg\max_\theta p(\theta \mid \theta_0, D_\tau^S). \tag{3}$$

A full derivation can be found in appendix A.2. The quality of the estimate of the adapted parameters is, however, limited by the finiteness of the support data. In fact, while it is unbiased (appendix A.5), it may suffer from high variance.

## 2.3 Variance Reduction

In this section we discuss the variance problem related to meta-learning and techniques to reduce it.

Assuming the support set is composed of $N$ i.i.d. samples from the task, the variance of the estimated adapted parameters can be expressed as follows:

$$\text{Var}\left[\hat{\theta}\right] = \frac{1}{N^2} \sum_{i=1}^N \text{Var}\left[\hat{\theta}_i\right], \tag{4}$$

which in the case of task overlap is non-zero. In fig. 4 we depict the variance of this estimate for a simple sinusoidal regression task, together with a comparison to our proposed variance-reduced estimation described in section 3. As can be seen, for standard GBML, the variance of the estimated parameters cannot get below the variance induced by the finite sampling of the support data. On the other hand, LAVA's estimation has a decreasing variance as training progresses.

To address the problem of high variance, we leverage upon a general method for reducing the variance of a sum of random variables (Hartung et al., 2011). Let $\theta_1, \ldots, \theta_N$ be a set of random variables with the same mean $\theta^*$ and different covariance $\Sigma_i$ for each $i = 1, \ldots, N$. Let $\hat{\theta} = \sum_{i=1}^N W_i \theta_i$ denote a weighted average with weights $W_i \in \mathbb{R}^{d \times d}$. We wish to estimate $W_i^*$ that yield a $\hat{\theta}$ with the minimum variance. We can define this variance reduction problem as:

$$\min_W \quad \text{Var}\left[\sum_{i=1}^N W_i \theta_i\right], \quad \text{subject to} \quad \sum_{i=1}^N W_i = I. \tag{5}$$

**Proposition 1** (Variance Reduction). *The solution to eq. (5) is given by:*

$$W_i^* = \left[\sum_{i=1}^N \Sigma_i^{-1}\right]^{-1} \Sigma_i^{-1}. \tag{6}$$

We provide a proof in appendix A.6 for completeness.

## 3 Method

In this section, we describe a novel method to model the estimation of the task-adapted parameters more accurately in the task overlap regime. Given a finite set of support points $D^S$, we want to approximate the optimal posterior of eq. (3). Let $p(\theta|x_i, y_i)$ denote the posterior w.r.t to one data point. Given the conditions of task overlap, (definition 1), this induces a distribution in parameter space. Another way of interpreting this posterior is that each data point provides evidence of what the possible true function could be. The full posterior $p(\theta|D^S)$ will thus lie at the *intersection* of all marginal posteriors. Given the i.i.d. assumption, this implies that:

$$p\left(\theta|D^S\right) \propto \prod_{i=1}^N p(\theta|x_i, y_i). \tag{7}$$

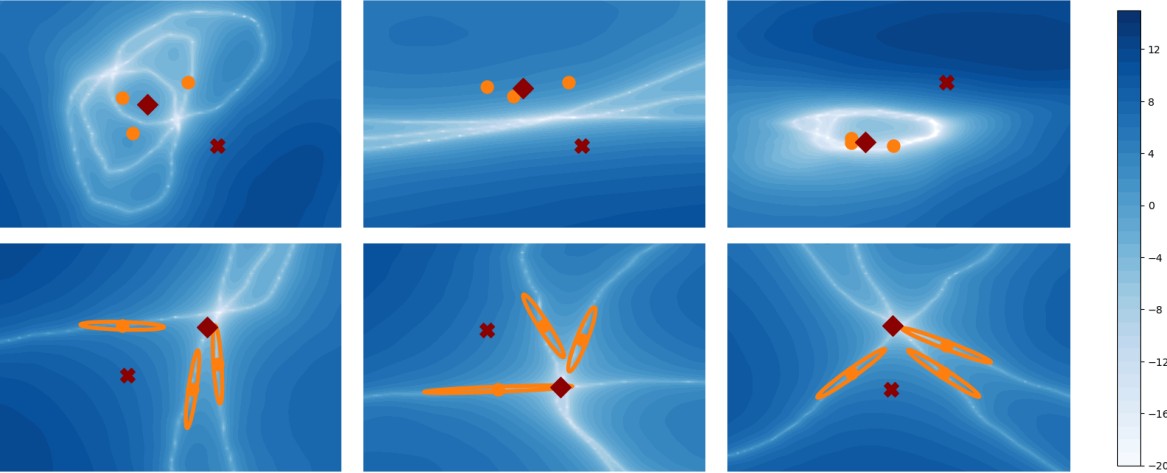

Figure 2: **The space of task-parameters adapts to the Hessian**. Sum of the logarithm of the loss for 3 support points over different parameters for the sine experiment. Values increase from white to dark blue. The red cross and the red diamond indicate the prior and the posterior, orange points are the single task-adapted parameters. *Top row:* Results for CAVIA. *Bottom row:* Results for LAVA, included is also the covariance for each support point.

A derivation can be found in appendix A.4. We propose to model the probability of the adapted parameters given each support point as a Gaussian distribution, $\mathcal{N}(\hat{\theta}_i, \Sigma_i)$, by means of Laplace's approximation (Kass et al., 1991). In particular, we define the maximum a posteriori (MAP) estimate to be the GBML original formulation i.e., $\hat{\theta}_i = \theta_0 - \alpha \nabla_{\theta_0} \mathcal{L}(\theta_0, D_i^S)$. The variance of the adaptation can then be defined as the inverse of the Hessian of the loss function evaluated in the adapted parameters i.e., $\Sigma_i = H_i^{-1}$. This approximation allows us to rewrite the estimate of the overall adapted parameters in eq. (7) as the product of these Gaussians. The most probable set of parameters given the provided support data will thus be the mean of the resulting Gaussian distribution:

$$\hat{\theta} = \left( \sum_{i=1}^{N} H_i \right)^{-1} \sum_{i=1}^{N} H_i \hat{\theta}_i. \tag{8}$$

One interpretation of GBML methods such as MAML (Finn & Levine, 2018) is that the posterior estimate assumes an equal covariance across all marginal posteriors, which in turn implies that the MAP estimate equals the average of the parameters. In fact, in the case of no task overlap, eq. (8) becomes equivalent to eq. (2). This follows from the fact that we perform *a single gradient step* to approximate $\arg\max_\theta p(\theta|D^S)$. Due to linearity of the gradient operator, $\arg\max_\theta p(\theta|D^S)$ corresponds to the average of $\{\arg\max_\theta p(\theta|x_i, y_i)\}_{i=1}^{N}$. However, in the case of anisotropic uncertainty in the adapted parameters, as in the task overlap regime, this leads to a sub-optimal estimation. On the other hand, when the Laplace approximation faithfully describes the underlying distribution, the estimate proposed in eq. (8) corresponds to the minimum variance estimator. This can be seen from proposition 1, where $\Sigma_i = H_i^{-1}$. The final assumption to make the results coincide is that the expectation over the adapted parameters is the same, i.e. $\mathbb{E}_{p(\tau)}[\hat{\theta}_i] = \theta^*$ for all $i = 1, \dots, N$. This $\theta^*$ can be seen as the optimal for the given task but is not necessary for the proof.

Compared to simple parameter averaging defined in eq. (4), we achieve a lower variance for any support set $D^S \sim p(\tau)$. Thus, on expectation over all tasks, we achieve a variance-reduced estimate.

The overall training procedure follows from GBML methods. We optimize the set of parameters $\theta_0$ through the objective of eq. (1) as a bi-level optimization procedure where $\theta_\tau$ is now defined according to eq. (8). Differently from GBML, the Laplace approximation allows for reshaping of the parameter space of the learned model. Embedding this new adaptation process in the optimization allows for a precise approximation. To minimize the loss, the model has to shape the parameter space such that, locally, the Laplace approximation can exactly estimate the posterior. This imposes certain constraints on the structure of the parameter space

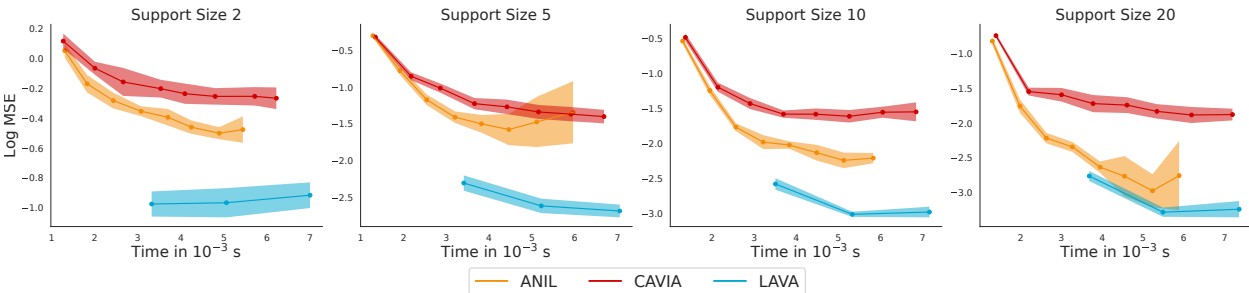

Figure 3: **Evaluation of Computation Time**: We evaluate the performance of the different methods (measured in terms of MSE) as a function of the computational time (in seconds) across different support sizes. The results show that, considering the same execution time (x-axis), LAVA outperforms both CAVIA and ANIL, with lower MSE. This result holds across all evaluated support sizes.

arising from task overlap. This learned parameter space can be precisely shaped to allow for an accurate Laplace Approximation due to the flexible nature of parametric neural networks. Consider once again the sine wave regression example. Each support point gives evidence for a space of possible solutions. When considering a model with a parameter space of dimension 2, the space of solutions in the adapted parameters is a one-dimensional curve. The optimal posterior corresponds to an intersection of the curves induced by each support point. In contrast to other GBML methods, LAVA allows for straightening out the region of parameters that minimize the inner loss function. This in turn allows for shaping of the parameter space that enables a precise Laplace Approximation. We illustrate this in fig. 2.

## 3.1 Computational Implementation

A limitation of the described method lies in an increased time complexity. Computing the Hessian on each single support point can severely affect the training time of methods that already require complex second-order calculations like in GBML, especially for over-parameterized neural networks. To minimize the computational burden, we consider performing adaptation in a subspace of the parameter space. In most of the experiments, we opt for the implementation of ANIL (Raghu et al., 2020), which performs adaptation over the last layer of a neural network as they argue that most of the adaptation takes place in the final layers. This allows us to compute the Hessian in closed form, drastically reducing the computational cost and keeping it independent of the dimensionality of the input space.

Nevertheless, the Hessian computation can result in an increase in computational time and LAVA is, in fact, more expensive than the standard GBML model. However, this computational complexity increase is paired with stronger performances. LAVA provides a more effective adaptation technique as one of its main features is the efficient use of the limited information given by the support. In this regard, LAVA provides a better trade-off between performances and computational complexity. To better analyze this trade-off we compared LAVA's performances against two GBML baselines in the Sine regression experiment by varying the number of inner loop adaptation steps. As shown in fig. 3, LAVA has a computational complexity comparable to CAVIA (Zintgraf et al., 2019) with 2 inner loop gradient steps and ANIL with 3. In appendix A.8 we provide a description of how to compute the Hessian when the loss is in the form of eq. (1).

Computing second-order derivatives, however, is known to be numerically unstable for neural networks, (Martens, 2016). We found that a simple regularization considerably stabilizes the training. Following Warton (2008), we take a weighted average between the computed Hessian and an identity matrix before aggregating the posteriors. For all of our experiments, we substitute each Hessian $H_i$ in eq. (8) with the following:

$$\tilde{H}_i = \frac{1}{1+\epsilon} \left( H_i + \epsilon I \right), \tag{9}$$

where $\epsilon$ is a scalar value and $I$ is the identity matrix of the same dimensionality of $H_i$. In our experiments, we consider $\epsilon = 0.1$.

## 4    Related Work

**Gradient-Based Meta-Learning** methods were first introduced with MAML (Finn et al., 2017a) and then expanded into many variants. Among these, Meta-SGD (Li et al., 2017) includes the learning rate as a meta parameter to modulate the adaptation process, Reptile (Nichol & Schulman, 2018) gets rid of the inner gradient and approximates the gradient descent step to the first-order. In CAVIA (Zintgraf et al., 2019), the adaptation is performed over a set of conditioning parameters of the base learner rather than on the entire parameter space of the network. Other works instead make use of a meta-learned preconditioning matrix in various forms to improve the expressivity of the inner optimization step  (Lee & Choi, 2018; Park & Oliva, 2019; Flennerhag et al., 2020).

**Bayesian Meta-Learning** formulates meta-learning as learning a prior over model parameters. Most of the work in this direction is concerned with the approximation of the intractable integral resulting from the marginalization of the task parameters. This has been attempted using a second-order Laplace approximation of the distribution (Grant et al., 2018), variational methods (Nguyen et al., 2020), MCMC methods (Yoon et al., 2018), or Diffusion models (Pavasovic et al., 2023). While these Bayesian models can provide a better trade-off between the posterior distribution of the task-adapted parameters and the likelihood of the data  (Chen & Chen, 2022), they require approximating the full posterior and marginalizing over it. LLAMA (Grant et al., 2018) proposes to approximate the integral around the MAP estimate through the Laplace Approximation on integrals. Given one gradient step, the posterior estimate is still performed by averaging. As we have shown, the averaging fails to account for inter-dependencies between the support points arising from task overlap.

**Model-based Meta-Learning** relies on using another model for learning the adaptation. One such approach is HyperNetwork (Ha et al., 2022), which learns a separate model to directly map the entire support data to the task-adapted parameters of the base learner. In Gordon et al. (2019), this is implemented using amortized inference, while in Kirchmeyer et al. (2022), the task-adapted parameters are context to the base learner. Alternatively, the HyperNetwork can be used to define a distribution of candidate functions using the few-shot adaptation data (Garnelo et al., 2018b;a) and additionally extend it using an attention module (Kim et al., 2019). Lastly, memory modules can be iteratively used to store information about similar seen tasks (Santoro et al., 2016) or to define a new optimization process for task-adapted parameters (Ravi & Larochelle, 2017). All of these methods can potentially solve the aggregation of information problem implicitly as the support data are processed concurrently. However, the learned model is not model-agnostic and introduces additional parameters.

## 5    Experiments

To begin with, we test the validity of using the Laplace approximation to compute the task-adapted parameters for a simple sine regression problem. Additionally, we show how LAVA exhibits a much lower variance in the posterior estimation in comparison to standard GBML. In appendix A.5, we demonstrate the unbiasedness of GBML and evaluate the noise robustness of GBML and LAVA against a model-based approach.

We further evaluate our proposed model on dynamical systems tasks of varying complexity in regard to the family of functions and dimensionality of the task space as well as regression of two real-world datasets. We compare the results of our model against other GBML models. In particular, our baselines include ANIL (Raghu et al., 2020), CAVIA, (Zintgraf et al., 2019) as a context-based GBML method, LLAMA (Grant et al., 2018), VFML and VR-MAML (Wang et al., 2021; Yang & Kwok, 2022) as a variance-reduced meta-learning methods, and MetaMix (Chen et al., 2021) as a meta-data augmented method. Experimental details are given in the appendix A.1.

### 5.1    Sine Wave Regression

To develop a further intuition of our model, we conduct experiments on the sine-wave regression problem introduced in Finn & Levine (2018). We investigate two aspects of our model. The first is in regards to the geometry of the learned parameters space (appendix A.7) and how well the Laplace approximation

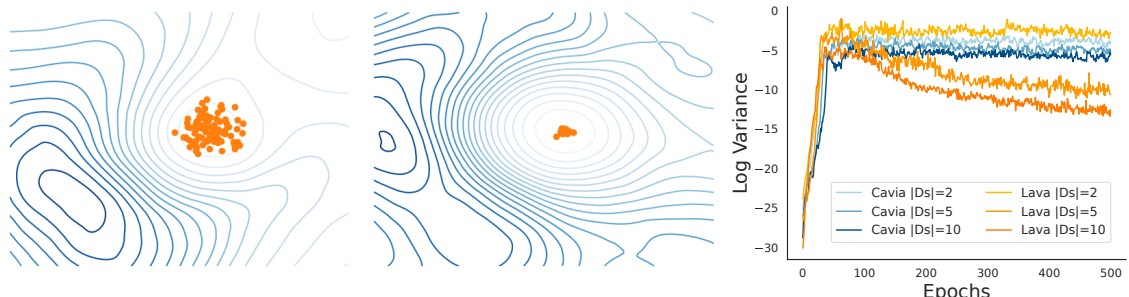

Figure 4: **Estimator's variance.** Variance of the task-adapted parameters given the same task but different support data points. *Left:* For CAVIA. *Center:* For LAVA. *Right:* Log variance of the distribution of the adapted parameters during training.

can accurately capture distributions in parameter space. The other aspect we investigate is measuring the variance reduction in our estimate. In the sine experiment, the dimensionality of the true task parameters is two, allowing us to visualize the learned parameter space. To this end, we train both LAVA and GBML with a conditional model (akin to CAVIA (Zintgraf et al., 2019)) on the sine-wave regression problem with a context vector of dimension two.

Given a single $(x, y)$ tuple, there exists a one-dimensional space of sine waves that pass through that point. This makes the aggregation challenging and thus allows us to test the benefits of approximating this subspace with the Laplace Approximation.

**Laplace Approximation Assumption** The first ablation aims at testing the quality of the Laplace approximation in modeling the distribution of the task parameters given each single data point. After the models have converged, we visualize the loss landscape induced by different support data points when using different task-adapted parameters. In particular, we sample a support dataset of three $(x, y)$ tuples and compute the logarithm of the mean squared error between the prediction of both models and the true $y$ sampled from a grid of tasks. The idea is that each data point's loss is minimized by a continuous space of parameters. When summing up the losses of these support points, in fact, very well-defined valleys can be noticed in the loss landscape. We illustrate the distribution of this sum of losses for a grid of task-adapted parameters in fig. 2 as a heat map for CAVIA (top row) and LAVA (bottom row). Additionally, the prior context (red cross), the single task adapted parameters (orange dots) and the aggregated final posterior (red diamond) are also shown. For our method, we provide a visualization of the Hessian matrix for each single task-adapted parameter.

**Variance Estimation** For the second ablation, we evaluate the variance of the posterior estimate for both CAVIA and LAVA using the sine regression framework. The variance of the estimator describes the difference between the task-adapted parameters from the sampling of different support sets from the same task. For this experiment, we fix the sine task and sample 100 different support sets each with 10 $(x, y)$ tuples. For each of these support points, we compute the resulting task-adapted parameters. We depict the spread of these distributions in fig. 4 (left and center) for CAVIA and LAVA respectively. Note that the scale of the parameter space as well as the inner learning rate is the same for the two methods. The right of fig. 4 illustrates the log variance of the task-adapted parameters during training for different support sizes. As can be seen in the figure, at the beginning of training, the variance increases suddenly as the models learn to use these parameters to solve the meta-learning problem. However, while CAVIA's context parameters variance remains high during the rest of the training, our method learns to reduce it consistently.

## 5.2 Differential Equations

We evaluate further on a set of complex regression tasks in the form of dynamical systems prediction. We consider 5 sets of Ordinary Differential Equations (ODEs):

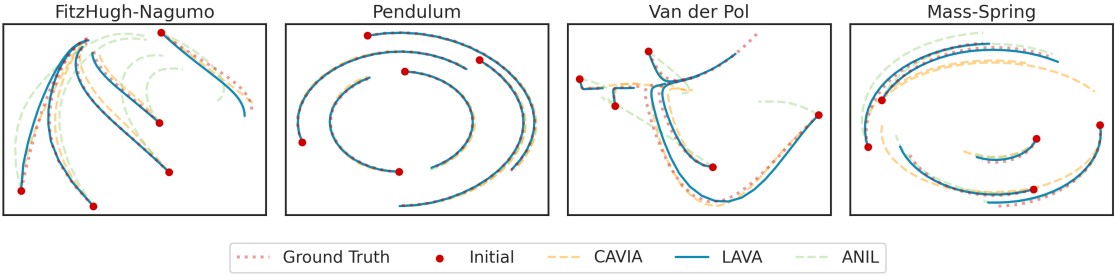

Figure 5: **ODEs qualitative results.** Rollout Trajectories for the dynamical systems' prediction with CAVIA, LAVA and ANIL. We consider the dynamics of systems with random parameters for 5 initial conditions. LAVA (blue line) is the only model that consistently predicts the evolution of the system (red dotted line).

- **FitzHugh-Nagumo**: A model for excitable systems, e.q. modeling the spike of a neuron.

- **Mass-Spring System**: A classic mass-spring dynamics model for prediction of the effects of gravity upon a mass connected to a spring.

- **Pendulum**: Describes the motion of an object of mass $m$ connected to a rodd of length $L$ suspended from a pivot.

- **Van Der Pol Oscillator**: An oscillating system that undergoes non-linear damping, determined by the parameter $\mu$.

- **Cartpole**: An inverse dynamics problem of an actuated cartpole with varying mass.

For each system, we consider samples from its vector field as our target data. We train all models for 300 epochs and compare their MSE on reconstructing the vector field, we present the results in table 1. For CAVIA and ANIL, we perform a grid search over up to 8 inner optimization steps and select the best-performing one. For a single gradient step, LAVA outperforms both CAVIA and ANIL even with a large number of adaptation steps. This increase in performance can be attributed to LAVA's use of second-order information through the curvature of the loss landscape. Further details about the dataset construction and choice of parameters are given in appendix A.1.

To further evaluate the top-performing models, we visualize roll-out trajectories devised from integrating the learned vector field in fig. 5. We can note that LAVA provides strong roll-out prediction in almost all cases.

Table 1: **MSE** ↓ by Dynamical System and Model for Support Size 10, including Cartpole results

| Model Name | FitzHugh -Nagumo | Mass-Spring ($\times 10^{-2}$) | Pendulum ($\times 10^{-2}$) | Van der Pol | Cartpole |
|---|---|---|---|---|---|
| CAVIA | $0.19 \pm 0.05$ | $1.00 \pm 1.36$ | $0.34 \pm 0.11$ | $3.57 \pm 1.18$ | $1.14 \pm 0.07$ |
| ANIL | $0.64 \pm 0.25$ | $0.09 \pm 0.04$ | $0.22 \pm 0.07$ | $12.29 \pm 6.84$ | $1.81 \pm 0.57$ |
| LLAMA | $1.18 \pm 0.57$ | $0.51 \pm 0.05$ | $3.29 \pm 0.37$ | $6.78 \pm 1.10$ | $1.77 \pm 0.32$ |
| VFML | $1.27 \pm 0.25$ | $0.53 \pm 0.07$ | $4.62 \pm 0.81$ | $6.36 \pm 1.34$ | $1.86 \pm 0.62$ |
| Metamix | $3.56 \pm 1.21$ | $0.65 \pm 0.11$ | $3.68 \pm 0.85$ | $46.01 \pm 8.29$ | $4.49 \pm 0.15$ |
| VR | $3.88 \pm 0.95$ | $3.28 \pm 0.28$ | $21.55 \pm 8.52$ | $32.26 \pm 6.42$ | $2.34 \pm 0.46$ |
| LAVA | $\mathbf{0.17 \pm 0.05}$ | $\mathbf{0.02 \pm 0.00}$ | $\mathbf{0.07 \pm 0.01}$ | $\mathbf{1.50 \pm 0.87}$ | $\mathbf{1.02 \pm 0.25}$ |

### 5.3 Real World Datasets

In the next experiment, we evaluate our model on two real-world datasets. The first is a regression task on the Beijing Air Quality Dataset (Zhang et al., 2017) while in the second we consider regressing a real-world

and synthetic Frequency-modulated radio signal subject to noise in the RadioML 2018.01A dataset (O'Shea et al., 2018).

**Air Quality:** We consider the Beijing Air Quality Dataset (Zhang et al., 2017) which is a time-series dataset containing recordings of air quality across 12 monitoring sites. The dataset contains hourly measurements during the time period from 01-03-2014 to 28-2-2017. We consider each monitoring site a separate task. Each task can be viewed as an interpolation task. A random contiguous subsequence of size $n_s + n_q$ is selected which is then split randomly into a support- and query-set. For each measurement, a time-variable $t$ is appended, indicating a positional embedding in the time-series. The task is then predicting the air-quality measurement from the given time $t$. We present the MSE in the left-most column of table 2. In this task, LAVA clearly outperform the baselines.

**RadioML:** Lastly, we evaluate the performances of LAVA compared with the baselines on regressing frequency-modulated radio signals. To do so, we make use of the RadioML 2018.01A dataset described in O'Shea et al. (2018). These are both synthetic and real-world radio signals recorded over-the-air and subject to noise. We devise the task as being able to regress each signal given a few data point recordings. For testing, we withhold 25% of the signals. We present the MSE in table 2 on the right-most column. Regression of these signals poses a considerable challenge for meta-learning methods. The change in frequency between signals exacerbates the task overlap conditions of the task. The results highlight this, as LAVA is the only method able to correctly regress the signals.

Table 2: **MSE** ↓ of model-predictions on the Air-Quality dataset and RadioML dataset.

| Model Name | Air-Quality $(\times 10^{-2})$ | RadioML |
|---|---|---|
| ANIL | $7.63 \pm 0.31$ | $0.047 \pm 0.0139$ |
| CAVIA | $8.25 \pm 0.16$ | $0.5016 \pm 0.0$ |
| VFML | $9.21 \pm 0.46$ | $0.156 \pm 0.1351$ |
| LLAMA | $15.37 \pm 7.36$ | $0.5017 \pm 0.0$ |
| Metamix | $56.76 \pm 13.52$ | $0.0471 \pm 0.0018$ |
| VR | NaN | $0.491 \pm 0.0081$ |
| LAVA | $\mathbf{4.65 \pm 0.30}$ | $\mathbf{0.0038 \pm 0.0002}$ |

## 6 Discussion and Conclusion

In this paper, we characterized the problem of *task overlap* for under-determined inference frameworks. We have shown how this is a cause of high variance in the posterior parameters estimate for GBML models. In this regard, we have proposed LAVA, a novel method to address this issue that generalizes the original formulation of the adaptation step in GBML. In particular, the task-adapted parameters are reformulated as the average of the gradient step of each single support point weighted by the inverse of the Hessian of the negative log-likelihood. This formulation follows from the Laplace approximation of every single posterior given by each support data point, resulting in the posterior being the mean of a product of Gaussian distributions. Empirically we have shown how our proposed adaptation process suffers from a much lower variance and overall increased performance for a number of experiments.

For regression tasks, the assumption of task overlap is of particular interest. On the other hand, classification tasks are inherently discrete and do not suffer from the problem of task overlap to the same extent as regression-like problems. Nonetheless, for completion, we provide in appendix A.10 the results of a few-shot classification experiment on *mini-Imagenet*. In this experiment, LAVA performs similarly to the other GBML baselines, confirming the different nature of the overall problem. However, the discrete nature of classification problems presents an avenue for future work in the possibility of incorporating adaptation over a categorical distribution of parameters. A second limitation is the computational burden of computing the Hessian. As described in section 3.1, and further discussed in appendix A.8, we overcome this limitation by restricting the adapted parameters to the last layer only (akin to ANIL (Raghu et al., 2020)). This allows us to compute

the Hessian in closed form. We show that the time complexity of our model is comparable to 2 inner steps of CAVIA and 3 inner steps of ANIL while showcasing superior performance.

An interesting extension to the proposed method would be to explore techniques to approximate the Hessian. This would allow us to extend the adapted parameters to the whole model. Possible approximations could include the Fisher information matrix or the Kronecker-factored approximate curvature (Martens & Grosse, 2015) to estimate the covariance of the Laplace approximation. Alternatively, it might be interesting to explore the direction of fully learning this covariance by following an approach similar to model-based methods.

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

# A  Appendix

## A.1  Experimental Details

For all of the experiments and all the baselines, we fix the architecture of the meta-learner $f_\theta$ to be a multi-layer perceptron with 3 hidden layers of 64 hidden units together with ReLU activations. We use a meta batch size of 10 tasks and train all the models with Adam (Kingma & Ba, 2014) optimizer with a learning rate of $10^{-3}$. We use the inner learning rate $\alpha = 0.1$ for the adaptation step and MSE as the adaptation loss. All experiments were run for 5 different seeds to compute mean and standard deviations. For LLAMA we use $\eta = 10^{-6}$, for PLATIPUS we scale the KL loss by 0.1, for BMAML we use 10 particles and use MSE rather than the chaser loss for a fair comparison. Other experiment-specific details include:

### A.1.1  Differential Equations

- **FitzHugh-Nagumo**:

$$\frac{du}{dt} = c(u - u^3/3 + v),$$
$$\frac{dv}{dt} = -\frac{1}{c}(u - a + bv),$$

  with $a, b, c \sim \mathcal{U}[0.1, 2.0]$. The initial positions are sampled as $u, v \in \mathcal{U}[-2.5, 2.5]$.

- **Mass-Spring**:

$$\frac{dx}{dt} = -\frac{\dot{x}}{m},$$
$$\frac{d\dot{x}}{dt} = -kx,$$

  with mass $m$ and spring constant $k$ sampled from $\mathcal{U}[0.5, 1.5]$ with initial conditions $x, \dot{x} \sim \mathcal{U}[-1, 1]$.

- **Pendulum**: The variables modeled is the angle of the pendulum and its angular velocity. The dynamics are described as:

$$\frac{d\theta}{dt} = \frac{\dot{\theta}}{ml^2},$$
$$\frac{d\dot{\theta}}{dt} = -mgl\sin(\theta).$$

  We sample mass $m$, length $l$ and gravity $g$ as $\mathcal{U}[0.5, 1.5]$ and initial conditions $\theta \sim \mathcal{U}[-\frac{\pi}{2}, \frac{\pi}{2}]$ and $\dot{\theta} \sim \mathcal{U}[-1, 1]$.

- **Van-Der Pol Oscillator** We can describe through a first-order ODE as:

$$\frac{dx}{dt} = y,$$
$$\frac{dy}{dt} = \mu(1 - x^2)y - x.$$

  We let $\mu \sim \mathcal{U}[0.1, 5.0]$ and $x, y \sim \mathcal{U}[-3, 3]$.

- **Cartpole:** The dynamics of a rigid body can be described as follows:

$$M(q)\ddot{q} + C(q, \dot{q})\dot{q} + g(q) = Bu.$$

  Here, $q$ is a generalized coordinate vector, $M$ is a matrix describing the mass, $C$ is a force matrix and $g(q)$ is the gravity vector. The system takes external force as input contained in $u$ which gets mapped into general forces by the matrix $B$. The inverse dynamics task involves predicting the control inputs $u$ given a target trajectory $\{q(s)\}$. We generate trajectories of an actuated cartpole with varying mass $M$ and train on the inverse-dynamics task.

### A.1.2 RadioML

The RadioML 2018.01A dataset (O'Shea et al., 2018) consists of measurements of over-the-air radio communications signals. The original dataset consists of signals with different modulation techniques and varying signal-to-noise (SNR) ratios. For the experiment presented in the paper, we restrict the dataset to signals with frequency modulation and an SNR greater or equal to 20 dB. Each signal is composed of two channels describing the In-phase and Quadrature components of the signal (I/Q). Here we consider the first 100 time-steps of each of these signals as the curve to be regressed.

### A.2 Bayesian View on GBML

GBML can be formulated as a probabilistic inference problem from an empirical Bayes perspective (Grant et al., 2018). The objective of GBML involves inferring a set of meta parameters $\theta_0$ that maximize the likelihood of the data $\mathbf{D} = \bigcup_\tau D_\tau$ for all tasks. Keeping the notation above and marginalizing over the task-adapted parameters, the GBML inference problem can be written as follows:

$$\theta_0 = \arg\max_\theta p(\mathbf{D}|\theta) = \arg\max_\theta \prod_\tau \int p(D_\tau^Q|\theta_\tau)p(\theta_\tau|D_\tau^S,\theta)d\theta_\tau, \tag{10}$$

where $p(D_\tau^Q|\theta_\tau)$ corresponds to the likelihood of each task's data given the adapted parameters and $p(\theta_\tau|D_\tau^S,\theta)$ is the posterior probability of the task-adapted parameters.

The integral in eq. (10) can be simplified by considering a *maximum a posteriori* (MAP) estimate of the posterior:

$$\theta_\tau^* = \arg\max_{\theta_\tau} p(\theta_\tau|D_\tau^S,\theta).$$

This simplifies the intractable integral to the following optimization problem:

$$\theta_0 = \arg\max_\theta \prod_\tau p(D_\tau^Q|\theta_\tau = \theta_\tau^*).$$

### A.3 Variance of parameters

Here we provide a short account of the variance of the GBML parameters estimator as reported in eq. (4):

$$\begin{aligned}
\mathrm{Var}\left[\hat{\theta}\right] &= \mathrm{Var}\left[\theta_0 - \alpha\nabla_{\theta_0}\frac{1}{N}\sum_{i=1}^N \mathcal{L}(\theta_0, D_i^S)\right] \\
&= \mathrm{Var}\left[\frac{1}{N}\sum_{i=1}^N \theta_0 - \alpha\nabla_{\theta_0}\mathcal{L}(\theta_0, D_i^S)\right] \\
&= \frac{1}{N^2}\sum_{i=1}^N \mathrm{Var}\left[\hat{\theta}_i\right].
\end{aligned}$$

### A.4 Intersection of the Posteriors

We show that the posterior over the parameters conditioned on the support set is proportional to the intersection of the posteriors given each single point in the support. The main idea here is applying the Bayes rule twice and making use of the i.i.d. assumption on the support data.

$$\begin{aligned}
p\left(\theta \mid D^S\right) &= \frac{p(D^S \mid \theta)p(\theta)}{p(D^S)} \\
&= \frac{\left(\prod_{i=1}^{N} p(x_i, y_i \mid \theta)\right) p(\theta)}{p(D^S)} \\
&= \frac{\left(\prod_{i=1}^{N} \frac{p(\theta|x_i,y_i)p(x_i,y_i)}{p(\theta)}\right) p(\theta)}{p(D^S)} \\
&= \frac{\prod_{i=1}^{N} p(\theta \mid x_i, y_i)}{p(\theta)^{N-1}} \\
&\propto \prod_{i=1}^{N} p(\theta \mid x_i, y_i).
\end{aligned}$$

### A.5 Gradient-Based Meta-Learning is an Unbiased Estimator

Here, we show that GBML with one gradient step is an unbiased estimator. Define the loss for one task as:

$$\mathcal{L}(\theta, \tau) = \mathbb{E}_{x \sim p(x|\tau)}\left[\mathcal{L}(\theta, x)\right].$$

Then the gradient w.r.t $\theta$ is an unbiased estimator:

$$\mathbb{E}_{x \sim p(x|\tau)}\left[\nabla_\theta \mathcal{L}(\theta, x)\right] = \nabla_\theta \mathbb{E}_{x \sim p(x|\tau)}\left[\mathcal{L}(\theta, x)\right] = \nabla_\theta \mathcal{L}(\theta, \tau).$$

Moreover, we measure empirically the bias of GBML and LAVA estimators. As a comparison, we include a fully learned network implemented as a HyperNetwork (Ha et al., 2022) that takes as input the entire support dataset and outputs the adapted parameters directly. Both the adaptation and the aggregation are learned end-to-end together with the downstream task.

We train these three models until convergence on the sine regression dataset. Then, we measure their performance on each task corrupted by Gaussian noise with a standard deviation of 3 on the support labels. The experiment is designed to test how the performance changes when increasing the support size. We show the difference in the loss between adaptation with and without noise for the three models and for different support sizes in fig. 6. Thus, we are effectively testing the ability of these estimators to recover the performances of the noiseless adaptation. Ideally, an unbiased estimator converges to the correct posterior with enough samples as long as the noise has zero mean. As can be seen in the figure, GBML methods are much more robust to these kinds of perturbations, while learned networks are not unbiased.

### A.6 Variance reduction

Below we give an account of the proof of proposition 1. Consider the variance reduction problem defined in eq. (5)

$$\min_{W} \quad \mathrm{Var}\left[\sum_{i=1}^{n} W_i \theta_i\right], \quad subject\ to \quad \sum_{i=1}^{n} W_i = I. \tag{11}$$

We have that

$$\mathrm{Var}\left[\sum_{i=1}^{n} W_i \theta_i\right] = \sum_{i=1}^{n} W_i \Sigma_i W_i^T.$$

By introducing a Lagrange multiplier $\lambda$, we reach the following optimization problem:

$$\min_{W} \quad F(\theta, W),$$

$$\mathrm{with} \quad F(\theta, W) = \sum_{i=1}^{n} W_i \Sigma_i W_i^T + \lambda \left(\sum_{i=1}^{n} W_i - I\right).$$

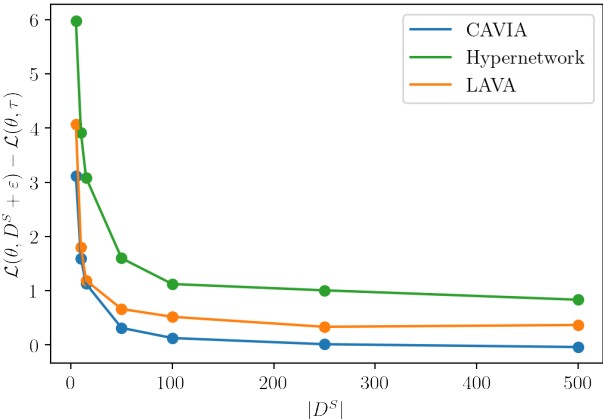

Figure 6: **GBML is an unbiased estimator.** Scaled MSE when adding noise to the support labels for the sine experiment with increasing support size.

By taking the derivative w.r.t $W_i$ and using the fact that $\Sigma_i$ is symmetric we find that

$$\frac{dF}{dW_i} = 2W_i\Sigma_i - \lambda.$$

Setting this equal to 0, we get

$$W_i = \frac{\lambda}{2}\Sigma_i^{-1}. \tag{12}$$

From the condition defined in eq. (11), we have

$$\frac{\lambda}{2}\sum_{i=1}^{n}\Sigma_i^{-1} = I,$$

$$\lambda = 2\left[\sum_{i=1}^{n}\Sigma_i^{-1}\right]^{-1}.$$

Plugging this into eq. (12), it follows that

$$W_i = \left[\sum_{i=1}^{n}\Sigma_i^{-1}\right]^{-1}\Sigma_i^{-1}.$$

Given these weights, $W_i$, the distribution of $\sum_{i=1}^{n}W_i\theta_i$ follows a normal distribution equivalent to the estimate in eq. (8). $\square$

## A.7  Geometry of the Parameter Space

In order to minimize the loss described in eq. (1) using LAVA's adaptation (eq. (8)), we implicitly make an assumption over the geometry of the task space. In this section, we expand on this assumption and provide an example of the limitations of using Gaussians to represent the distribution of solutions in parameter space. In fact, the model has to shape the parameter space such that, locally, the Laplace approximation can exactly estimate the posterior. This imposes certain constraints on the structure of the parameter space arising from task overlap.

From the assumption of task overlap (definition 1), we have that $f(\cdot, x) : \mathcal{T} \to \mathcal{Y}$ is non-injective, or that for each $(x, y) \in \mathcal{X} \times \mathcal{Y}$, the pre-image $\mathcal{M}_{x,y} = f_x^{-1}(y)$ defines a subspace of the task-manifold and $\bigcup_{x,y} \mathcal{M}_{x,y}$

---

**Algorithm 1** LAVA Pseudo-Code

---

**Require:** $p(\mathcal{T})$ distribution of tasks
**Require:** $\alpha, \eta, \epsilon$ hyperparameters.
**Ensure:** Output results

1: Randomly initialize $\theta_0$
2: **while** not done **do**
3:     Sample batch of tasks $\mathcal{T} \sim p(\mathcal{T})$
4:     **for all** $\tau \in \mathcal{T}$ **do**
5:         Sample $D_\tau^S, D_\tau^Q \sim \tau$
6:         **for all** $(x_i, y_i) \in D_\tau^S$ **do**
7:             Evaluate $\hat{\theta}_i = \theta_0 - \alpha \nabla_\theta \mathcal{L}(\theta_0, x_i, y_i)$
8:             Evaluate $H_i = \frac{d^2}{d\theta^2} \mathcal{L}(\hat{\theta}_i, x_i, y_i)$
9:             Evaluate $\tilde{H}_i = \frac{1}{1+\epsilon} (H_i + \epsilon I)$
10:         **end for**
11:         Evaluate $\tilde{H} = \sum_i \tilde{H}_i$
12:         Evaluate $\hat{\theta}_\tau = \tilde{H}^{-1} \sum_i \tilde{H}_i \hat{\theta}_i$
13:     **end for**
14:     Update $\theta_0 = \theta_0 - \eta \nabla_{\theta_0} \sum_{\tau \in \mathcal{T}} \mathcal{L}(\hat{\theta}_\tau, D_\tau^Q)$ using each $D_\tau^Q$
15: **end while**

---

defines a *cover* of the task space $\mathcal{T}$. As the Laplace approximation estimates the space with a normal distribution, we require the underlying fiber $\mathcal{M}_{x,y}$ to be simply connected, or in other words, each element $(x, y)$ induces a path-connected space of model parameterizations in which every path can be continuously deformed into another one.

An example of a space of tasks with a non-trivial topology could be an annulus or a disk with a hole inside it. Such a space can arise in problems of goal-oriented navigation. Consider an agent in a 2-dimensional plane $P$ where the task provides the control inputs to reach a given goal position specified by a coordinate $(x, y) \in P$. Given trajectories provided by expert demonstrations, infer the goal-conditioned policy. In this setting, each support point corresponds to a position of the agent, and the loss is defined as the $L_2$ distance to the goal. If the goal can be situated anywhere on the plane $P$, then a single support point $(x_s, y_s)$ defines a space of possible goal positions as a circle around the current position. By continuity, this implies that the space of possible parameters $M_{x_s, y_s}$ that yield the possible policies must be non-simply connected as well. This poses problems for the Laplace approximation, as the Gaussian distribution implicitly holds an assumption on the topological properties.

### A.8 A Note on Computational Complexity

Here we provide a description of how we compute The Hessian in closed form with respect to the loss defined in eq. (1) and how to implement it in practice on a standard automatic differentiation framework.

Assuming the dimensions on the output to be: $\mathcal{Y} \subseteq \mathbb{R}^{k \times 1}$. In the case of a multi-layer perception, define $f_\psi \in \mathbb{R}^{d \times 1}$ the network up to the last layer excluded and augment it with an appropriate one-padding to vectorize the bias i.e., $z = \begin{bmatrix} f_\psi(x) \\ 1 \end{bmatrix} \in \mathbb{R}^{(d+1) \times 1}$. Moreover, define the adaptation parameters to be the weights and the bias of the last layer i.e., $\theta = [W, b] \in \mathbb{R}^{k \times (d+1)}$. We can rewrite the loss of eq. (1) for one data point of the support as:

$$\mathcal{L}(\theta, x_i, y_i) = \|\theta \cdot z - y\|_2^2.$$

The corresponding Hessian can be written as:

$$H_i = 2 I_{k \times k} \otimes (z \cdot z^T).$$

---

**Algorithm 2** Hessian Pseudo-Code in PyTorch.

---

**Require:** $x_i, y_i, f_\psi, W_\theta, b_\theta$

1: $z = f_\psi(x_i)$
2: $H_W = I_{k \times k} \otimes (z \cdot z^T)$
3: $H_b = I_{k \times k} \otimes z$
4: $H = 2 \begin{bmatrix} H_W & H_b \\ H_b^T & I_{k \times k} \end{bmatrix}$

---

The notation on the most standard automatic differentiation frameworks like PyTorch (Paszke et al., 2019) differs somehow from the above way of computing the Hessian. In Pseudo-code algorithm 2 we provide the necessary code for computing the Hessian.

### A.9 Additional Experiments

### A.9.1 Condition Number

We measure the condition number of the Hessian for LAVA under both the conditional setting and when only updating the final layer. We calculate the condition number $\kappa(H)$ for $H = \sum_{i=1}^{N} H_i$ and $\tilde{H}$ calculated similarly with the approximate Hessians defined in eq. (9). We present the results across different support sizes in table 3 below.

| | Condition Number $\kappa(H)$ | | Regularized Condition Number $\kappa(\tilde{H})$ | |
| --- | --- | --- | --- | --- |
| Model | Epoch=1 | Epoch=200 | Epoch=1 | Epoch=200 |
| LAVA + CAVIA (dim=2) | $4.96 \times 10^0$ | $1.94 \times 10^0$ | $1.04 \times 10^0$ | $1.96 \times 10^0$ |
| LAVA + CAVIA (dim=4) | $1.31 \times 10^2$ | $6.26 \times 10^2$ | $1.04 \times 10^0$ | $41.03 \times 10^0$ |
| LAVA + CAVIA (dim=8) | $5.28 \times 10^6$ | $1.78 \times 10^6$ | $1.03 \times 10^0$ | $66.73 \times 10^0$ |
| LAVA + CAVIA (dim=16) | $1.35 \times 10^{10}$ | $5.31 \times 10^9$ | $1.04 \times 10^0$ | $36.00 \times 10^0$ |
| LAVA + ANIL | $1.89 \times 10^{11}$ | $1.36 \times 10^{11}$ | $74.36 \times 10^0$ | $217.86 \times 10^0$ |

Table 3: Condition and Approximate Condition Numbers for Support Size 10

The condition number increases with the number of adaptation parameters increases. A high condition number signifies that the matrix is close to being singular, meaning it is difficult to invert. In the context of matrix inversion, this implies that small errors in the input data can lead to disproportionately large errors in the output, making the inversion process highly sensitive and potentially unreliable. In the table, we also present the condition number at convergence. This approximate condition number is derived from the approximate Hessian, as defined in eq. (9). Initially, the regularization promotes an even distribution of singular values, but as training progresses, the matrices become increasingly skewed. This behavior is expected, as the Hessian for each support point $H_i$ should be elongated in specific directions, resulting in a relatively high condition number.

### A.9.2 Additional Sine Results

Here we provide additional results for the sine regression experiment. Using the same experimental settings described in section 5.1 and appendix A.1, we present MSE and standard deviations for 5 seeds for LAVA and baselines in table 4. In particular, we provide results for support size equal to 1 as an ablation. The MSE value is noticeably higher as one support point is not sufficient to identify the underlying task accurately. Nevertheless, results show comparable performances between LAVA and ANIL when $|D^S| = 1$ as eq. (8) becomes equivalent to the adaptation described in eq. (2). Qualitative results comparing the effect of different support points and the addition of Gaussian noise on the support are shown in fig. 8. The figure shows how the prior of the model (Green dotted line) evolves into the posterior for LAVA (orange curve) and ANIL (blue curve), this is compared with the ground truth curve (green dashed line). The effects of task overlap

| Models | $|D^S| = 1$ | $|D^S| = 2$ | $|D^S| = 10$ | $|D^S| = 20$ |
|---|---|---|---|---|
| ANIL | $171.07 \pm 10.19$ | $46.05 \pm 6.42$ | $1.32 \pm 0.1$ | $0.37 \pm 0.02$ |
| CAVIA | $247.6 \pm 4.02$ | $56.75 \pm 2.5$ | $2.96 \pm 0.46$ | $1.44 \pm 0.23$ |
| VFML | $286.0 \pm 0.95$ | $106.31 \pm 7.14$ | $31.72 \pm 4.21$ | $18.63 \pm 1.8$ |
| LLAMA | $248.48 \pm 4.12$ | $109.75 \pm 12.41$ | $29.08 \pm 4.81$ | $18.31 \pm 2.01$ |
| Metamix | $329.9 \pm 13.53$ | $120.12 \pm 7.44$ | $42.04 \pm 4.27$ | $24.85 \pm 2.25$ |
| VR | $318.21 \pm 17.05$ | $338.3 \pm 22.91$ | $283.05 \pm 21.38$ | $292.28 \pm 32.98$ |
| LAVA | $\mathbf{169.38 \pm 9.18}$ | $\mathbf{7.01 \pm 2.02}$ | $\mathbf{0.11 \pm 0.02}$ | $\mathbf{0.09 \pm 0.02}$ |

Table 4: Model performance MSE ($10^{-2}$) based on different support sizes on the Sine wave regression problem.

can be seen in the plots in the left column. The support size is small, GBML models like ANIL manage to fit the support points (red crosses) but fail to identify the true underlying distribution.

## A.10  Mini-Imagenet

| Model | 5-ways 1-shot | 5-ways 5-shot |
|---|---|---|
| ANIL | $45.94 \pm 0.94$ | $62.86 \pm 0.26$ |
| CAVIA | $47.84 \pm 0.41$ | $63.09 \pm 0.51$ |
| LLAMA | $40.19 \pm 0.85$ | $56.50 \pm 0.15$ |
| VFML | $49.60 \pm 0.5$ | $\mathbf{66.20 \pm 0.80}$ |
| Metamix | $\mathbf{50.51 \pm 0.86}$ | $65.73 \pm 0.72$ |
| VR | $49.20 \pm 1.40$ | $63.60 \pm 0.80$ |
| LAVA | $46.69 \pm 1.45$ | $61.51 \pm 0.97$ |

Table 5: Results Mini-Imagenet with support sizes 1 and 5

We further experiment with classification on the Mini-Imagenet dataset (Vinyals et al., 2016). We use the training-set split as used in Ravi & Larochelle (2017) which leaves 64 classes for training, 16 for validation and 20 for test. We experiment with 5-way classification in either a 1-shot or 5-shot setting. We train the models for 1000 epochs and perform model selection by choosing the one with the best performance on the validation set. We present results on the test set in table 5.

Standard classification benchmarks such as Mini-Imagenet test the capability of the model to incorporate high-dimensional data in the form of images. Some of the methods are optimized toward image data and attempt to efficiently learn a well-structured representation space of images, such that the adaptation reduces to modifying decision boundaries. In particular, few-shot image classification problems in this form are inherently discrete problems that do not suffer as extensively from the task overlap assumption as outlined in definition 1. In this context, the Gaussian assumption on the parameter distribution is not an accurate one, as each element of the support induces a *multi-modal* distribution in the parameter space. Thus variance reduction becomes inaccurate, and unbiased methods such as ANIL prove better. Nevertheless, LAVA manages to get competitive performances on this benchmark as well.

## A.11  ODEs Computational Complexity

We provide additional computational complexity analysis for the ODEs experiments. As it can be noticed in fig. 7, for most of the experiments LAVA provides a comparative advantage in terms of computational complexity and general performances.

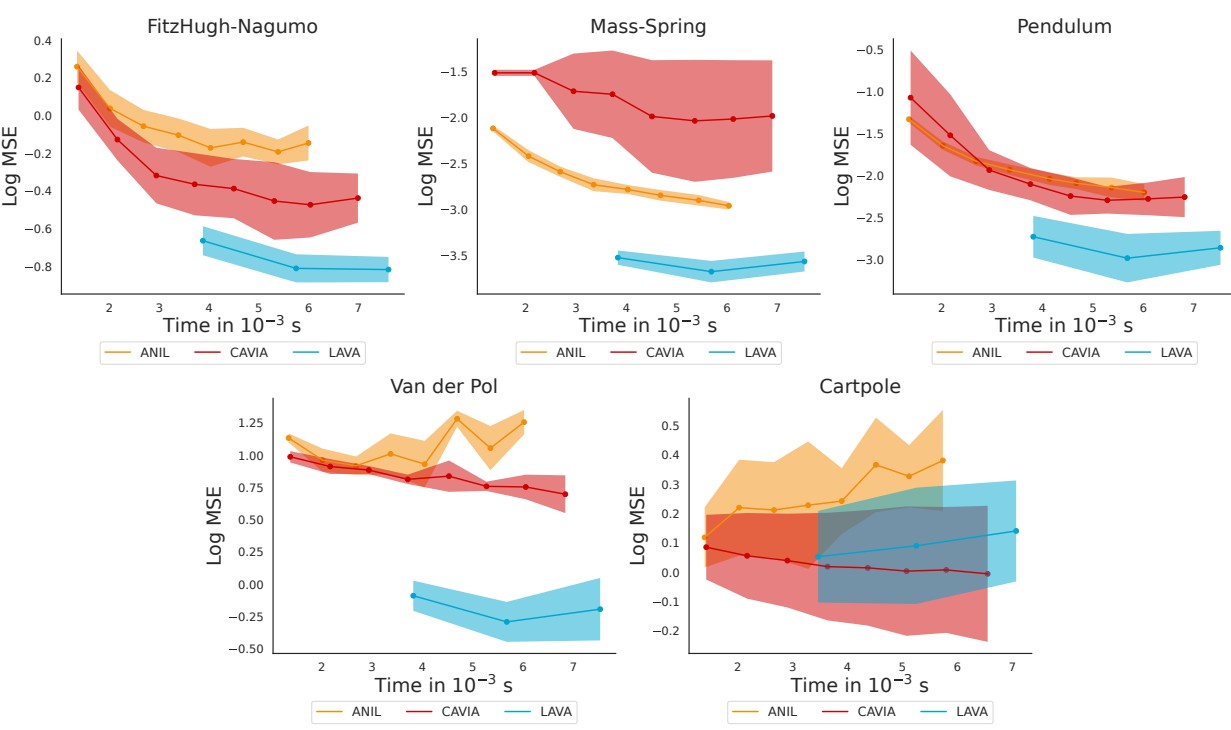

Figure 7: **Further Evaluation of Computational Time.** Computational Complexity results for ODEs regression with support size 5 and a different number of adaptation steps.

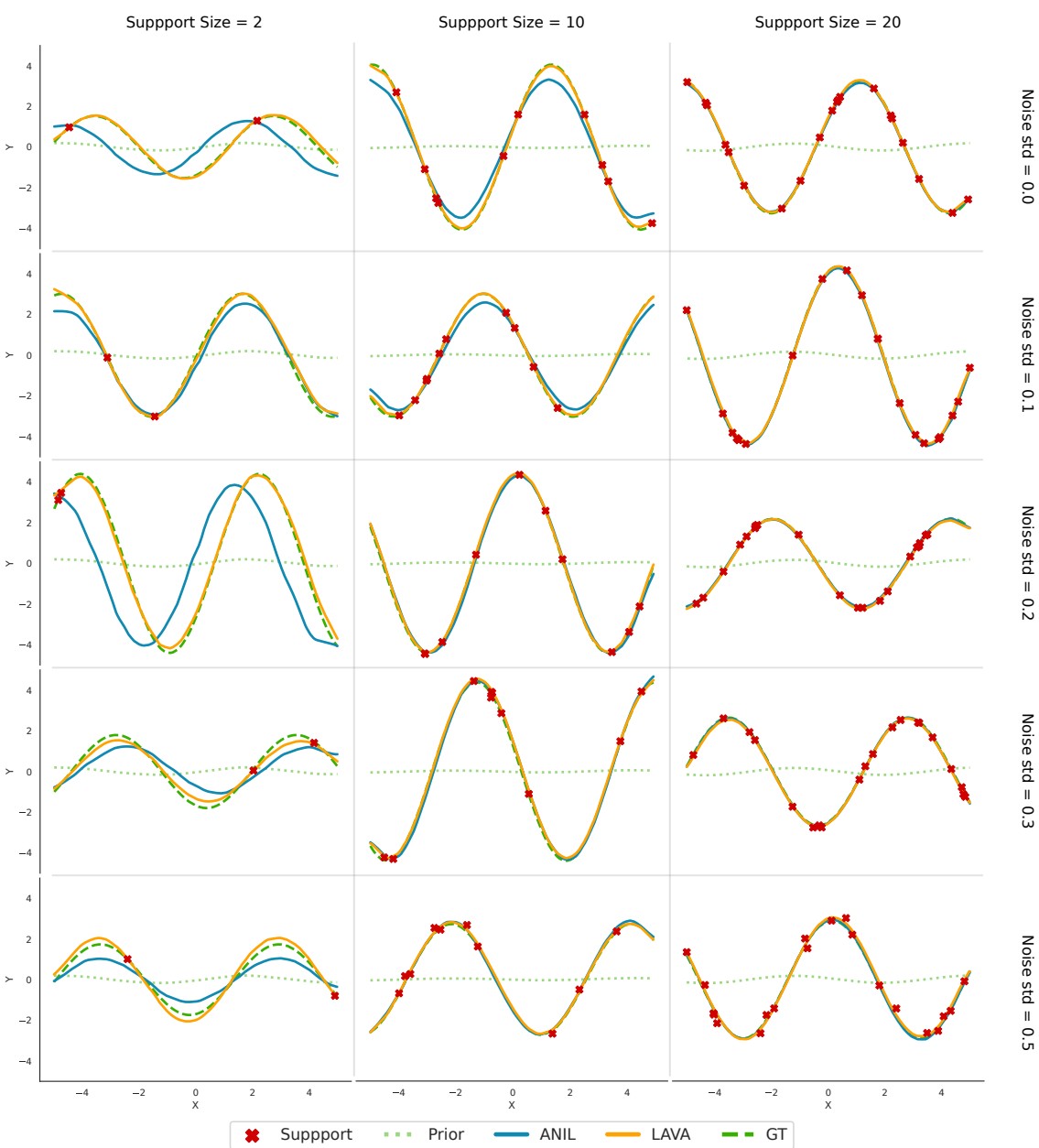

Figure 8: **Qualitative Results for Sine Regression.** Estimated sinusoidal curves with varying amounts of support size and noise in the support data compared with GBML.

