# OpenReview forum: "Reducing Variance in Meta-Learning via Laplace Approximation for Regression Tasks"
_TMLR — Accepted by TMLR_

### Review · Reviewer_cuAw · 2024-07-05

**Summary Of Contributions:**

This paper introduces Laplace Approximation for Variance-reduced Adaptation (LAVA) to address high variance in Gradient-Based Meta-learning (GBML), especially due to task overlap. LAVA reduces variance by weighing each support point based on the variance of its posterior, using the Laplace approximation and computing the Hessian of the loss function.

**Audience:**

Yes

**Claims And Evidence:**

No

**Requested Changes:**

# Corrections/comments
* The authors implicitly assume that there exists some true function parameters $\theta^*$. This should be putted into an assumption. In addition, I am unsure about the convexity of the objective. Thus, the authors should carefully make an assumption about that there exists some set of true parameters, $\Theta^*$.
* The notion of task overlap is presented as a property, i.e., Property 1. For me, this is clearly an assumption.
* The example after Property 1 is not properly defined. Having a running examples (i.e., the sine wave regression problem) is a good, but the authors should define its component properly.
* Sec. 2.2. change to equations 1 and 2. (Better: change to (1)-(2), e.g., see comment below about cleveref).
* Define how $\theta_{0}$ is learned.
* The Hessian is very expensive to compute. Why do the authors use some quasi method instead? The authors could explore the correlations and then maybe just use a quasi Hessian method which only update the diagonal. E.g., in the same flavour of using Adagrad instead of some quasi-Newton method to make the gradient steps.
* In the discussion about variance reduction I think the authors could explore more how bit the "mini-batch" should be to have reasonable variance reduction.
* In section 3.1 the authors discuss the computation costs of the Hessian, and propose to use some quasi-Hessian in (9). Why is the Hessian ill-posed? And, what is the $\epsilon$ used for this weighted aveaged?
* In the experiments section the LAVA methods seem to outperform all baselines in terms of epochs. I do not think it is a fair comparison to use second-order information, and thereby I think the experiments should be done in terms of computational costs.
* Correct the missing dots and comma in all equations.
* Don't define equations numbers on equations that are not referred to.
* Correct the references to equations, e.g., use the cleveref package. And then use (1) and not write Equation 1.
* Correct the format of the citations.

**Strengths And Weaknesses:**

I want to preface this section by stating that I am not an expert in this area (i.e., meta-learning) so I apologize for any inaccuracies.

# Strengths
* The notion of task overlap and addressing variance reduction by aggregation in GBML is good.
* The paper is overall well-written and clearly structured, with a lot of figures.

# Weaknesses
* I am not convinced about the theoretical claims in paper. The theoretical analysis is too heuristic, and the authors should be more carefull in their analysis. Some quantities are defined with words and thereby the analysis becomes too superficial.
* Despite efforts to mitigate it, the computation of the Hessian remains resource-intensive, which may limit the method's scalability to larger networks or datasets.

---

> ### Author Response · Authors · 2024-08-15
>
> We thank the reviewer for the comments and for the time spent reviewing our paper. We now address the weaknesses (W), and requested changes (R) pointed out by the reviewer:
> - **(W1, R1) Assumption of parameters $\theta^\*$:** We thank the reviewer for the proposed clarification. The proposed adaptation function described in Equation (8) (in the updated paper) is indeed not convex whenever the functional form of $f_\theta$ is general. In fact, using a neural network to parameterize $f_\theta$ does not allow for global optimality guarantees. We also kindly point out that the general form of Gradient-Based Meta-Learning adaptation described in Equation (2) (in the updated paper) also lacks convergence guarantees when used with non-linear function approximators as a neural network. Moreover, the empirical results provided in the paper suggest the feasibility of the proposed algorithm in practice. We included this assumption in Section 3 of the revised version of the paper.
> - **(W1, R2) Notion of task overlap:** We thank the reviewer for the suggestion. The notion of task overlap is introduced in the paper as a defining characteristic of some continuous meta-learning tasks (such as regression tasks) where the dimension of the task is greater than what a single support point provides. In the paper we specifically study these tasks and thus assume these abide to this definition. Nevertheless, LAVA does not require overlap of the task necessarily as it would naturally converge to standard GBML, as we discussed in Section 3. We have updated the paper to refer to Task overlap as a definition.
> - **(W1, R3) Definition of example after Property 1:** We thank the reviewer for pointing this out. We have expanded the example with a more precise definition in Section 2.1 of the updated paper.
> (W1, R4) Equation notation: We thank the reviewer for identifying the mistake, we have corrected it in the updated version of the paper.
> - **(W1, R5) Learning $\theta_0$:** $\theta_0$ is the only parameter which is learned in our model and is optimized through the objective defined in Equation (1). In our setting, the adaptation $\mathcal{A}$ in Equation (1) is defined as Equation (8), where $\hat{\theta_i}$ and $H_i$ both depend on $\theta_0$. When training the algorithm we minimize the loss In Equation (1) with respect to $\theta_0$ only. We made this connection more explicit in Section 3 the revised version of the paper.
> - **(W2, R6.1) Computational expense of computing the Hessian:** We agree with the reviewer on the increased computational complexity of the proposed method. To limit this computational burden we designed LAVA to only optimize the final layer of the network. This allows us to compute the Hessian in close form as described in Appendix A.8, sensibly reducing the computational cost. We highlight in Figure 3 that, with this design choice, LAVA overall provides a better trade-off in terms of computational complexity vs. performance, when compared with other Gradient-Based Meta-Learning approaches.
> - **(W2, R6.2) The use of a quasi-method:** We use the quasi-method to ensure numerical stability. We found empirically that the hessian matrix can be ill-conditioned at the start of training which requires numerical stability. Furthermore, we calculate the hessian over the parameters of the final layer of the network, which yields a closed-form solution. One could further disregard the correlation terms and only consider the diagonal. We have provided a further experiment in Appendix A.9 where we show the condition number of the Hessian with or without regularization at both the start and end of training.
> - **(R7) Variance with mini-batch size:** We have added additional experiments comparing the effects on the variance of the estimator in the sine experiment with different sizes of the adaptation dataset in Section 5.1. As expected the variance of the estimator is considerably smaller for LAVA for reasonable sizes of the dataset. Nonetheless, an increased amount of data does reduce the variance of the estimated set of parameters for GBML methods as well.

---

> > ### Author Response · Authors · 2024-08-15
> >
> > - **(R8) Why the Hessian is ill-posed + $\epsilon$ value used:** The Hessian is a measure of the curvature around a parameter $\theta$. In the case of task overlap, a single data point induces a valley in the parameter space, as there are several models which could fit this point. These valleys are visualized in Figure 2, where we plot the covariance matrix induced by the inverse Hessian as well. The analytical interpretation of these valleys is that the Hessian is ill-conditioned, that is, its largest singular value is strictly greater than its lowest. This induces errors in the numerical approximation of the inverse which is why we introduce a regularization term. We included another experiment where we measured the condition value of the Hessian for both the original and the approximation in Appendix A.9.1. From the results, we can see that the regularization effectively makes the Hessian well-posed at the start of training, which then is still “stretched out” as in Figure 2 at the end of training. In short, the regularization encourages a more numerically stable training. The epsilon we use is $\epsilon = 0.1$. We have made the reason for why the Hessian is ill-posed clearer in Appendix A.9.1 of the updated document.
> > - **(R9) Experiments in terms of computational costs:** We agree with the reviewer on the advantage of using second-order information. The main motivation for LAVA is that most GBML methods use the information provided in the adaptation set inefficiently. The results provided in the tables (1)-(2) show asymptotic performances of the compared methods. Moreover, LAVA shows a general advantage in terms of performance vs. computational time as shown in Figure 3. We have performed the same experiment on the computational time trade-off for the ODEs experiment, Appendix A.11. The new results confirm the comparative advantage that LAVA has with respect to standard GBML methods.
> > - **(R10-13) Corrections:** We thank the reviewer for pointing out these mistakes. In the updated version of the paper we have:
> >   - Corrected the missing dots and commas in all equations.
> >   - Removed equation numbers on equations that are not referred to.
> >   - Updated and corrected the references to equations using the cleveref package.
> >   - Updated the format of the citations.

---

### Review · Reviewer_vixc · 2024-07-18

**Summary Of Contributions:**

This paper mainly targets meta-learning. The authors point out that the previous gradient based meta-learning cannot well handle the ambiguity among tasks specified as task overlap. To this end, they propose a novel method which adopts Bayesian model-averaging to aggregate posterior information induced by each support sample. The proposed method shows strong performance on several different meta-learning benchmarks.

**Audience:**

Yes

**Claims And Evidence:**

Yes

**Requested Changes:**

Please refer to the weaknesses and provide more discussion.

**Strengths And Weaknesses:**

Strengths:
1. The method is clearly introduced.
2. The authors have presented extensive experiments.


Weaknesses:
1. I wonder if the authors can validate the choice of modelling parameter distribution as Gaussian.
2. The authors may discuss how the proposed method can improve the baseline gradient-based methods when support size is extremely limited, e.g. 1-shot.
3. I notice that the proposed method performs worse than ANIL on miniImageNet 5-shot experiment. Is there any proper reason for this result?
4. Fig.3 seems not so intuitive. The authors may update this figure for better understanding.
5. It seems the proposed method can decrease the standard deviation among different trials in some settings (e.g. Pendulum and Van der Pol in Tab.1) while also having large s.d. in Tab.4. More discussion on this would be better.

---

> ### Author Response · Authors · 2024-08-15
>
> We thank the reviewer for the comments and for the time spent reviewing our paper. We now address the weaknesses (W) pointed out by the reviewer:
> - **(W1) Modelling parameter distribution as Gaussian:** Modelling the parameter distribution with a Gaussian allows us to compute the posterior probability of the adapted parameters in closed-form by means of the Laplace approximation. As described in the last paragraph of Section 3 (before Section 3.1), the error in the approximation is reduced by embedding the approximation in the loss objective (see Figure 2). As discussed in Appendix A7 “Geometry of the Parameter Space”, this approximation is valid whenever the geometry of the task space can be represented by a unimodal distribution such as a Gaussian distribution (e.g. no holes in the task space). Empirically validating different distributions to represent the adapted parameters requires a different approximation method that would still allow for a closed-form solution of the maximum-a-posteriori estimation problem. This is an interesting future research direction, yet outside the scope of the current work. A discussion on the assumptions and conditions for the validity of our model is in Appendix A.9.
> - **(W2) Improvement of baseline GBML with LAVA:** We thank the reviewer for the comment, The 1-shot setting entails a particular situation where our method reduces to standard MAML. This can be seen in Equation (8) (in the updated paper). When $N=1$ the adaptation formulation reduces to Equation (2) (in the updated paper) as $\hat{\theta} = \hat{\theta_1}$. We have included an additional experiment evaluating the performance of our method with different amounts of support size in Appendix A.9.2. The results highlight that, as expected, with support size 1 the performance of our method becomes equivalent to standard GBML methods.
> - **(W3) miniImageNet results:** We would like to point out that this evaluation considers an out-of-scope scenario for our work - our contribution focuses on regression tasks. However, due to the fact that it is an important benchmark in Meta-learning, and to discuss the limitations of our method in classification tasks, we have presented and discussed the MiniImageNet results in Appendix A10. For this scenario, the Gaussian assumption on the parameter distribution is not an accurate one, as each element of the support induces a multi-modal distribution in the parameter space. Thus variance reduction becomes inaccurate, and unbiased methods such as ANIL prove better. However, surprisingly, our method still performs competitively with baseline methods, **despite not being designed to work with classification tasks**. We have clarified this point in Appendix A.10.
> - **(W4) Figure 3:** We have modified the figure and clarified the caption in the updated version of the paper. The figure describes the comparison in performances and computational time trade-off for LAVA and two baselines. By varying the number of adaptation steps we can increase the performances at the cost of higher computational complexity for any GBML method. The figure shows how LAVA, in general, provides a better tradeoff as its loss is below the baselines for the same computational execution time.
> - **(W5) miniImageNet standard deviation:** We thank the reviewer for the comment. As previously discussed, MiniImageNet is a classification task outside the scope of our work - we focus our contribution on meta-learning for regression tasks. We evaluated our method in this task to highlight the limitations of our method when applied in out-of-scope scenarios that violate our assumptions on the geometry of the task space. This, in turn, has a repercussion on the adaptation procedure in LAVA as the Hessian does not represent well the uncertainty in the estimated parameters. As such, the variance in performance is sensibly higher. We have clarified the out-of-scope nature of this evaluation in Appendix A.10 and highlight that, despite this, our method still performs competitively to the baseline methods.

---

### Review · Reviewer_TjnE · 2024-08-07

**Summary Of Contributions:**

This paper mainly focuses on the task overlap phenomenon in the context of gradient-based meta-learning. Generally, this is an interesting topic. As far as I know, few works have discussed this problem. In this paper, in order to solve the task overlap problem, Laplace Approximation for Variance-reduced Adaptation (LAVA) is proposed. Extensive empirical results reveal the effectiveness of the proposed LAVA method.

**Audience:**

Yes

**Claims And Evidence:**

Yes

**Requested Changes:**

1. Could you please provide more explanations about the Laplace approximation of the probability of the adapted parameters?

2. I feel confused about the definition of $\mu_i$ and $\Sigma_i$ in Section 3 of Page 4. It would be better to provide more justifications for the definition here.

3. The problem studied in the paper is very interesting. I also noticed that Pavasovic et al. [1] also studied this problem from the perspective of score matching. Could you please discuss LAVA and their work?

[1] Pavasovic et al. MARS: Meta-learning as score matching in the function space, ICLR 2023.

**Strengths And Weaknesses:**

__Pros__:
- The problem the paper focuses on is interesting and worth studying;
- The paper is well organized, and the visualizations are easy to read;
- Extensive empirical results demonstrate the effectiveness of the proposed LAVA method.

__Cons__:
- The definitions of the formulations are not well-organized, which makes the paper unreadable.

---

> ### Author Response · Authors · 2024-08-15
>
> We thank the reviewer for the comments and for the time spent reviewing our paper. We now address the weaknesses (W), and requested changes (R) pointed out by the reviewer:
> - **(W1) Organization of the paper.** We have improved the definitions throughout the paper, pointed out by the reviewer, in blue in the updated document.
> - **(R1) Explanations about Laplace Approximation.** Laplace’s approximation serves as a second-order approximation of the true posterior, which follows from the Taylor expansion of the posterior around its mode. Similar to the Taylor expansion, it serves as a correct approximation around the mode of the posterior. In the case of sine-wave regression, a single data point induces a posterior distribution with a flat mode (as there exists a line in parameter space representing possible sine-waves that fit this single data point). Thus, the Laplace approximation yields an elongated normal-distribution with variance defined through the Hessian. An intuition is that the Hessian provides a measure of the curvature in the parameter space, which in this case is flat in one direction. We have updated the discussion on the Laplace approximation in Section 2.3 of the updated document.
> - **(R2) Definitions of $\mu_i$ and $\Sigma_i$:** For a given support point $D_i = (x_i, y_i)$ we wish to model a distribution for the possible task-adapted parameters $p(\theta|x_i, y_i)$ it could yield. In this work we assume that the distribution of the task-adapted parameters induced by the support points follow a normal distribution: making use of the Laplace’s approximation, the mean is set to the $\text{argmax}_{\theta}$ of the likelihood, which in the case of one point is estimated to be the single-step gradient update. The variance is set as the inverse Hessian $H_i^{-1}$. This gives our $\mu_i$ and $\theta_i$ of our normal distribution for a single point. This follows from Laplace's approximation being a second-order approximation of the true posterior. Furthermore, it alleviates the computation of the full posterior $p(\theta|D_S)$ as the product of Gaussians. We have updated our discussion on the conditions for when such an approximation is accurate in Appendix A.7.
> - **(R3) Connections to MARS:** Thank you for the additional reference. MARS [1] presents a third way of performing meta-learning alongside GBML and model-based methods, such as hypernetworks, that we previously discussed. MARS instantiate meta-learning as only learning the prior distribution $p(f)$, which corresponds to the task-distribution $p(\mathcal{T})$ in our work. They learn this distribution through score-matching - similar to many other works that study functional diffusion processes. At test-time, they can then perform inference by plugging the pre-trained score-matching network in a functional stochastic variational gradient descent framework, to move from function space to parameter space and perform conditional sampling from the posterior $p(\theta|x,y)$. We would like to point out that this work differs from ours on a number of points:
>   + Our work specifically targets the problem of estimating argmax $p(\theta|x,y)$. Thus, we do not consider estimating the full distribution and are not bayesian in that sense.
>   + We consider the MAP estimate specifically when there are limited samples of $x$, $y$ in which uncertainty can arise. MARS fits each training task using a GP with a pre-specified kernel. This inherits all the problems of GP-estimation, which can be especially faulty in the low-data regime.
>   + At test time MARS assume that the test data $D^* = (X^*, y^*)$ overlaps with their samples from the measurement distribution $X \sim v(X)$. This follows from their use of a learnt score network $s_{\theta}(X, Y)$ acting on a fixed grid of data (obtained from the GP-estimation). This inherits problems similar to other model-based methods, which also act on sets of data using neural networks $A(X,Y) \to \theta$. One way to view it is that we specifically want to address the construction of an aggregation network $A(X,Y) \to \theta$ that is agnostic to any grid or sample size. The gradient-estimator is one such principled way, and has advantageous properties such as being unbiased. However, it can be improved, by increasing bias but subsequently reducing variance, as proposed in our method.
>
> We have added MARS to the related work section (Section 4) in the updated paper.

---

### Author Response · Authors · 2024-08-15
**Updated Paper**

We thank all the reviewers for taking the time to review our submission and for the comments. We have uploaded a revised paper with changes marked in blue.

---

### Decision · Action_Editor_1Wh3 · 2024-09-20

**Recommendation:** Accept as is

**Comment:**

All of the reviewers voted positively for this paper, citing the solid empirical analysis and sharp presentation in formulating their "overlap" problem of interest. I agree with them, and believe the paper can be accepted as-is (up to de-anonymization for the camera-ready version).

**Audience:**

Meta-learning is a very broad framework, and task overlap is a real issue that arises when designing meta-learning tasks with limited information. There should definitely be an audience for this paper.

**Claims And Evidence:**

The paper deals with gradient-based meta-learning methods, and offers a new model using a variance-based re-weighting scheme, where the variance is evaluated using a posterior that is Laplace-approximated. The overall formulation is presented clearly, and all the reviewers were satisfied with the empirical evidence provided by the authors for the efficacy of their method.